# A low-background Tet-On system based on post-transcriptional regulation using Csy4

**Yicheng Zhou, Chaoliang Lei, Zhihui Zhu** \* 

Hubei Insect Resources Utilization and Sustainable Pest Management Key Laboratory, College of Plant Science & Technology, Huazhong Agricultural University, Wuhan, Hubei, China

\* zhihui@mail.hzau.edu.cn

**Data Availability Statement:** All relevant data are within the manuscript and its Supporting Information files.

**Funding:** This study was supported by the National Natural Science Foundation of China (grant no.

## Abstract

On account of its stringent regulation and high rate of induction, the tetracycline regulatory system is used extensively for inducing target gene expression in eukaryotes. However, under certain circumstances, its associated background expression can be problematic, as in the expression of highly toxic proteins. We found that when using the Tet-On 3G system to drive expression of the *kid* toxin gene in sf9 insect cells, a higher percentage of cells were killed than when using an empty vector in the absence of the induction agent doxycycline, thereby indicating the leaky expression of this inducible expression system. Moreover, we found that the tetracycline-controlled transcriptional silencer (tTS) does not effectively reduce the background expression of the Tet-On 3G system in sf9 cells. However, Csy4, a Cas9 homologous protein in the CRISPR family with sequence-specific endonuclease activity, was found to be effective in reducing the Tet-On 3G system-associated background expression, although there was a concomitant reduction in the maximum induced expression. Nevertheless, we found that modification of the system via incorporation of TRE-controlled anti-sense *csy4* in combination with a WSSVie1 (Δ23) promotor-driven sense *csy4* significantly reduced the leaky expression of the Tet-On 3G system, and that the level of induction was higher than that initially obtained. This optimized Tet-On 3G system can significantly reduce cell death attributed to the background expression of Kid under uninduced conditions. Therefore, we developed a novel low-background inducible expression system for use in insect cells and potentially in other organisms including mammals based on post-transcriptional regulation using Csy4.

## Introduction

The tetracycline regulatory expression system is widely used to regulate gene expression and has been demonstrated to function in yeast [1], plants [2], insects [3, 4], and mammals [5, 6]. However, owing to the detrimental effects of Tet-On system-associated background expression, considerable attention has focused on measures that can be adopted to reduce such unwanted expression; for example, by screening rtTA protein mutants that are more sensitive to doxycycline via viral evolution, and achieving higher induction efficacy using lower concentrations of doxycycline [7]. Furthermore, by optimizing the TRE promoter, Loew et al. (2010)

31872290 and 31572329 to ZZ). The funders had no role in study design, data collection and analysis, decision to publish, or preparation of the manuscript.

**Competing interests:** The authors have declared that no competing interests exist.

obtained lower background expression of the Tet-On system and higher induction of target gene expression. Moreover, incorporating a tetracycline-controlled transcriptional silencer (tTS) in the Tet-On system has been demonstrated to be an effective measure for reducing the background expression level. The tTS is a fusion protein composed of TetR and the KRAB-AB silencing domain [8, 9], and in the absence of doxycycline, it can bind to the *tetO* sequence in the tet-response element (TRE) region and inhibit target gene expression. In the presence of doxycycline, the tTS dissociates from the TRE promoter, and subsequently, rtTA binds to *tetO* to promote expression of the target gene [10]. In mammalian systems, the combination of tTS and the Tet-On regulatory system has been demonstrated to have no appreciable effect on the ability of the system to induce transcription, and can significantly reduce the background expression level of the system [11, 12].

Kid, a toxin protein from the toxin–antitoxin system of the *Escherichia coli* R1 plasmid, can inhibit the proliferation of yeast, *Xenopus laevis*, and human cells, and also induce human cell apoptosis [13, 14]. The endoribonuclease Csy4 is a Cas9 homologous protein of the clustered regularly interspaced short palindromic repeat (CRISPR) system that can recognize and cleave specific RNA sequences [15]. Csy4 has multiple applications in molecular biology research, and has been used to expand the target range of the CRISPR/Cas9 system in zebrafish [16], induce the formation of circular RNA [17], and inhibit gene expression in transgenic plants [18]. By placing a Csy4 recognition sequence in the 5' untranslated region (5'UTR) of the mRNA, Csy4 can cleave mRNA to inhibit gene expression, whereas inserting a Csy4 recognition sequence at the 3' end of mRNA after Csy4 cleavage can maintain mRNA stability [19]. In this study, we examined the potential utility of Csy4 in the post-transcriptional regulation of inducible genes to control the background expression of highly toxic genes.

## Materials and methods

### Plasmid construction

All promoter, gene and polyA signal were obtained via polymerase chain reaction (PCR) using the primers listed in S1 Table. The *Tet-On 3G* gene and CMV promoter (Pcmv) were cloned from a pCMV-Tet3G vector (Clontech) and the TRE3G promoter and firefly *luciferase* gene were cloned from a pTRE 3G-Luc vector (Clontech). The *csy4* gene was amplified from a pSQT1601 plasmid, which was a generous gift from Keith Joung (Addgene plasmid # 53369). The *tTS* gene was cloned from pTet-tTS plasmid (Clontech). The *mcherry* and the *kid* gene (GenBank accession no. NC_022885.1) were cloned from pmcherry-kid plasmid. The White spot syndrome virus immediate-early 1 promoter (Pwssvie1) and the 23-bp truncated WSSV IE1 promoter (Pwssvie1 (Δ23)) were amplified from White spot syndrome virus (GenBank accession no. AF440570). The Orgyia pseudotsugata nucleopolyhedrosis virus (OpMNPV) immediate-early 2 promoter (Popie2), OpMNPV IE2 polyadenylation signal (opie2pA) and OpMNPV major envelope glycoprotein (GP64) polyadenylation signal (opgp64pA) were cloned from OpMNPV (GenBank accession no. NC_001875.2). The Autographa californica multiple nucleopolyhedrovirus (AcMNPV) immediate-early 1 (IE1) promoter (Pacie1) was cloned from AcMNPV (GenBank accession no. NC_001623). The Simian vacuolating virus 40 polyadenylation signal (sv40pA) and the Herpesvirus thymidine kinase polyadenylation signal (HSVTKpA) were cloned from pFastBac Dual plasmid (Invitrogen).

An mCherry expression cassette was constructed by inserting the WSSVIE1 promoter PCR fragment, *mCherry* gene, and opgp64 polyA signal into a pXL-BacII vector at *Bam*HI–*Spe*I sites. The resultant plasmid was designated plasmid 1 (p1, Fig 1A). Plasmid 2 was obtained by inserting a Popie2-kid-opie2pA expression cassette into plasmid 1 at *Bam*HI–*Xho*I sites (p2, Fig 1A), whereas plasmid 3 (p3, Fig 2A) was constructed by inserting a Popie2-Tet-On 3G-

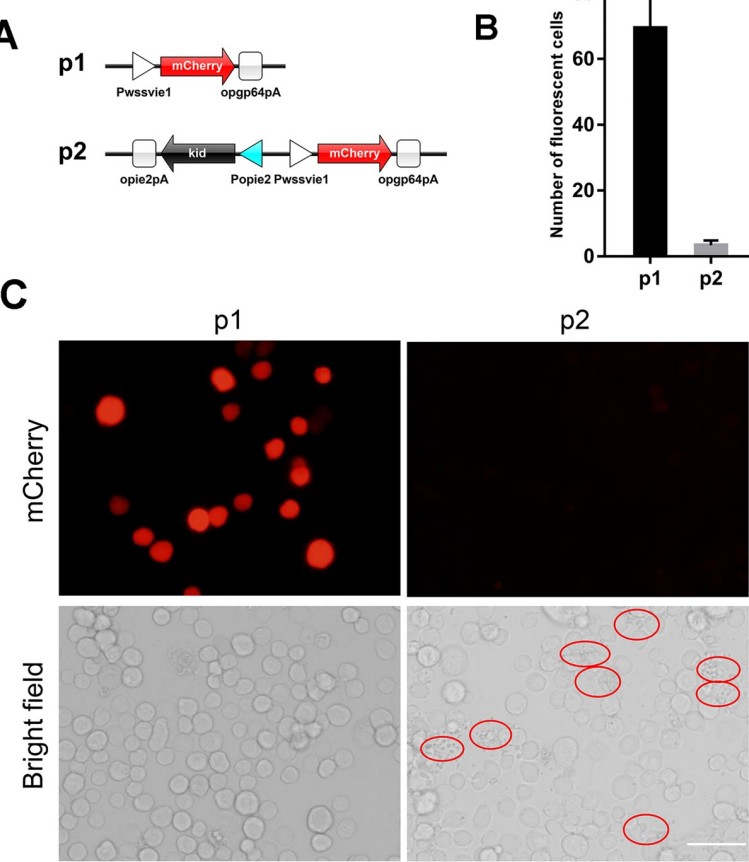

**Fig 1. Expression of Kid causes cell death in sf9 cells.** (A) A schematic diagram of vectors. Plasmid 1 (p1): the mcherry expression cassette was inserted into pXL-BacII vector. Plasmid 2 (p2): the kid expression cassette was added to plasmid 1. (B) Count of red fluorescence-positive cells in (C). Means and standard deviation were determined from three representative images (one from each of three independent biological replicates). (C) p1 and p2 were transfected into sf9 cells and the photographs were taken at 2 days post-transfection. The red-circled cells are fragmented and dead cells, illustrating the effects of Kid expression. Scale bar = 50 μm.

opie2pA expression cassette into plasmid 2 at *Bam*HI–*Xho*I sites and a TRE3G-kid-sv40pA expression cassette at *Nsi*I-*Xho*I sites. A Pacie1-tTS-HSVTKpA expression cassette was inserted into plasmid 3 at *Eco*RI–*Spe*I sites to produce plasmid 4 (p4, Fig 2A), and plasmid 5 (Fig 3A) was obtained by inserting the firefly *luciferase* gene into plasmid 3 to replace the *kid* gene at *Not*I–*Sal*I sites. Plasmid 6 (Fig 3A) was obtained by replacing the *kid* gene with *luciferase* gene at *Not*I–*Sal*I sites in plasmid 4. Plasmid 7 was obtained by inserting a csy4 recognition site between the TRE3G promoter and the *luciferase* gene by replacing the TRE3G promoter with TRE3G promoter and csy4 recognition site at *Nsi*I–*Not*I sites in plasmid 6, and insertion of the Pcmv-csy4-HSVTKpA expression cassette into this plasmid (Fig 3A). Replacement of the cmv promoter in plasmid 7 with a wssvie1 (Δ23) promoter produced plasmid 8 (p8, Fig 3A), whereas plasmid 9 was constructed by replacing the Pacie1-csy4-HSVTKpA expression cassette with a Pcmv-csy4-HSVTKpA expression cassette (p9, Fig 3A). Plasmid 10 (p10, Fig 3A) was generated by inserting a TRE3G-rcsy4 expression cassette at an *Eco*RI site in plasmid 8, whereas plasmid 11 (p11, Fig 4A) was obtained by replacing the *luciferase* gene with *kid* gene at *Not*I–*Sal*I sites in plasmid 10. All the sequences of the plasmids used in this study were listed in S1 File.

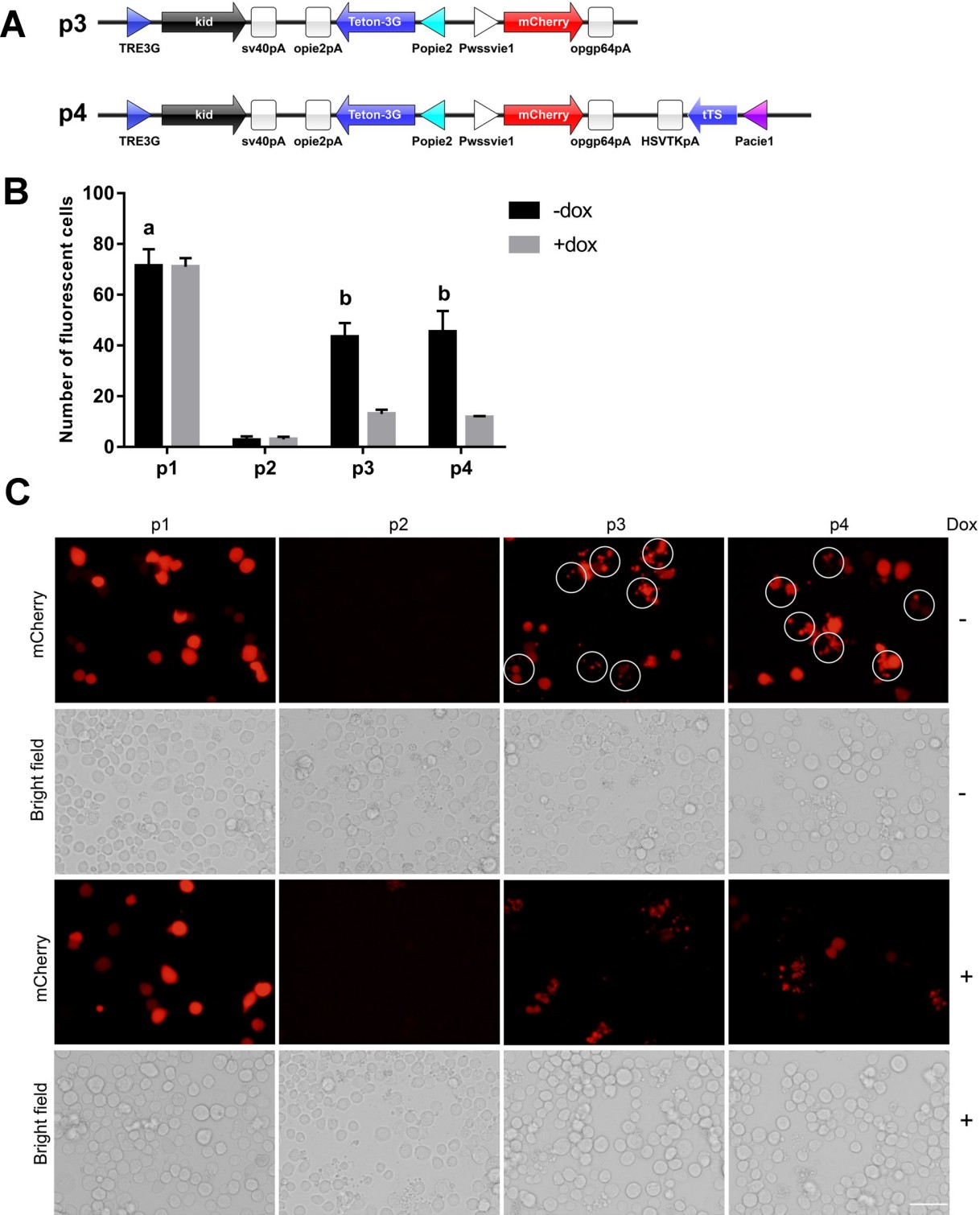

**Fig 2. Leaky expression of the Tet-On 3G and Tet-On 3G-tTS systems.** (A) A schematic diagram of vectors. Plasmid 3 (p3): the *kid* gene expression was controlled by TRE3G promoter control, and the Tet-On 3G transcriptional activator protein was controlled by the Opie2 promoter. Plasmid 4 (p4): Added the tTS gene expression cassette to the p3 plasmid. (B) Count of red fluorescence-positive cells in (C). Means and standard deviation were calculated from three representative pictures (one from each of three independent biological replicates). Values followed by different letters with a column are significantly different at P < 0.05 according to one-way ANOVA. (C) Sf9 cells were transfected with plasmid p1, p2, p3, or p4, and 6 h later, doxycycline (50ng/μL) was added to the medium as indicated. The fluorescence expression and

status of the insect cells were observed under a microscope at 48 h after transfection. The circled cells are fragmented and dead cells. Scale bar = 50 μm.

## Cell culture and transfection of the sf9 cell line

sf9 cells were maintained at 27˚C in TNM-FH medium (Hyclone) supplemented with 10% fetal bovine serum (Hyclone) and 1% penicillin/streptomycin (GIBCO). Plasmids for transfection were purified using a Plasmid Mini Kit (Omega), and concentrations were determined by using a NanoDrop 2000 spectrophotometer (Thermo). sf9 cells were seeded in 24-well plates

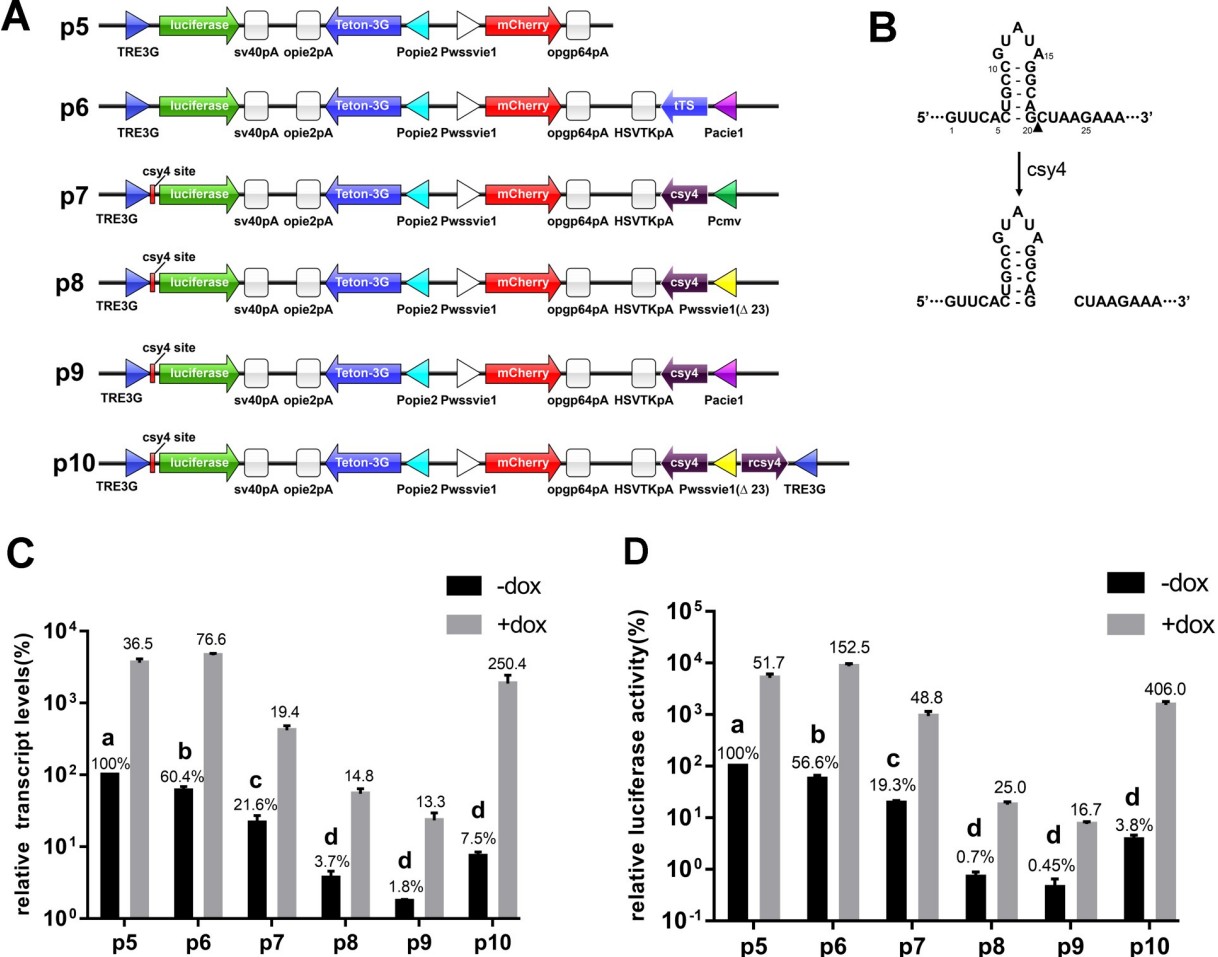

**Fig 3. Reduction in the background expression level of the Tet-On 3G system using the csy4 system.** (A) A schematic diagram of vectors. Plasmid 5 (p5): control plasmid, firefly luciferase expression was controlled by the Tet-On 3G system. Plasmid 6 (p6): contained an additional tTS expression cassette compared to plasmid 5. Plasmid 7 (p7): Added a 28 bp csy4 recognition sequence between the TRE3G promoter and the luciferase gene, and added a csy4 gene which was driven by CMV promoter. Plasmid 8 (p8) and plasmid 9 (p9) were similar to plasmid 7 expect that CMV promoter was replaced with WSSVie1(Δ23) or Acie1 promoter. Plasmid 10 (p10): Compared to plasmid 8, there was an additional reverse csy4 expression cassette controlled by TRE3G promoter. (B) The csy4 28-nucleotide substrate sequence and the fragment after digestion are shown above. The black triangle indicates the cleavage site. (C) RT-qPCR analysis for the change of firefly luciferase transcript at 48 h post-transfection with the indicated plasmids. The luciferase relative transcript level of plasmid 5-transfected cells in the absence of doxycycline was set at 100%. (D) Relative luciferase activity was measured and calculated at 48 h post-transfection with the indicated plasmids. The relative luciferase activity values of plasmid 5-transfected cells in the absence of doxycycline was set at 100%. The background expression level and fold induction are indicated above each column without and with doxycycline induction, respectively. Error bars indicate the standard deviation calculated from three independent experiments.

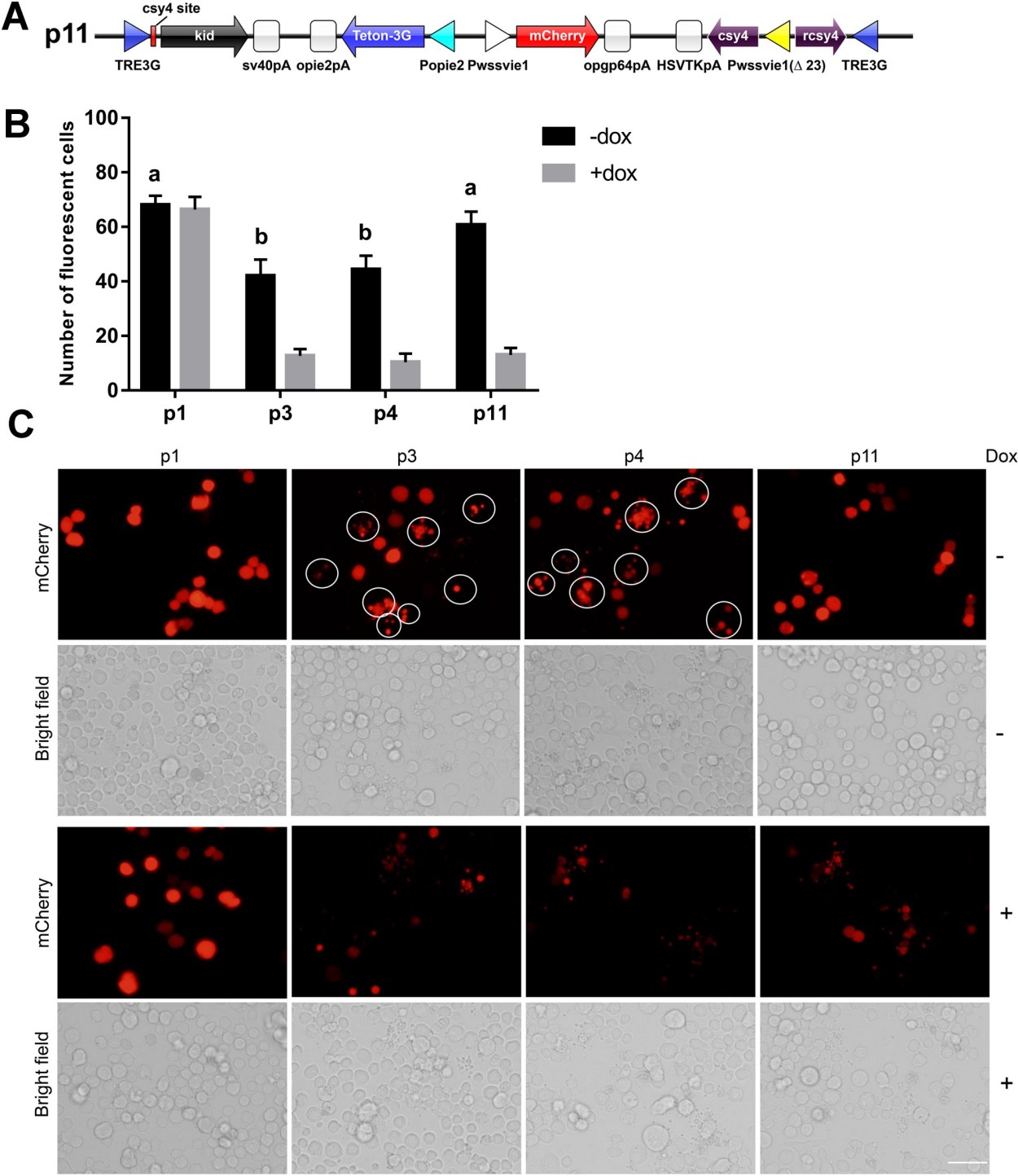

**Fig 4. The Tet-On 3G-Csy4 system tightly regulates *kid* gene expression.** (A) Vector construction diagram. Plasmid 11 (p11): The *luciferase* gene in plasmid p10 was replaced with the *kid* gene sequence. (B) Count of red fluorescence-positive cells in (C). Means and standard deviation were calculated from three representative images (one from each of three independent biological replicates). Values followed by different letters with a column are significantly different at $P < 0.05$ according to one-way ANOVA. (C) The fluorescence expression and cell growth were observed in sf9 cells at 48 h after transfection with p3, p4, or p11 in the absence or presence of doxycycline (50ng/μL). The circled cells are fragmented and dead cells. Scale bar = 50 μm.

and transfected with 1 μg plasmid using Cellfectin II reagent (Invitrogen) according to the manufacturer's protocol. All transfections were performed independently three times in triplicate.

### Fluorescence microscopy

sf9 cells were transfected with different plasmid as described in the previous section and were imaged at 48 h post-transfection with an Olympus IX71 imaging system using an RFP light cube or bright field.

### Luciferase reporter assays

In order to determine levels of Tet-On 3G system-associated background expression, we used a dual-luciferase system to normalize cell numbers and transfection efficiency. Cells were grown in the absence or presence of doxycycline (50 ng/mL) in 24-well plates. The vectors and internal reference plasmid pxl-BacII-Rluc, which can express *Renilla* luciferase in sf9 cells, were co-transfected into the cells using Cellfectin II reagent (Invitrogen) according to the manufacturer's protocol. Relative luminescence was measured at 48 h post-transfection using a Dual Luciferase Reporter assay kit (Transgen). Luminometric analysis was conducted using a PerkinElmer infinite 200 plate reader (Tecan).

### Real-Time quantitative Polymerase Chain Reaction (RT-qPCR)

RT-qPCR was performed to determine the transcript levels. Plasmid transfection was the same as luciferase analysis. Samples were collected 48 hours after transfection. The total RNA was isolated using Trizol regent (Takara) according to instructions of the manufacturer. The cDNA was synthesized from the total RNA using the cDNA Reverse Transcription Kit (Takara) in accordance with manufacturer's instructions. SYBR Green qPCR Master Mix (Transgen), specific primers of target firefly luciferase gene and internal control renilla luciferase genes were used in PCR reactions. PCR conditions were initial denaturation at 95°C for 30s, followed by 45 cycles at 95°C for 10 s and 60°C for 30 s. The CT values of firefly luciferase gene were normalized to the CT value of renilla luciferase gene and the relative fold expression changes were estimated by 2-[Delta][Delta] CT method.

### Statistical analysis

All experiments were performed in triplicate with three independent biological replicates, and the results were expressed as the mean ± standard deviation (SD). Data were analyzed using one-way ANOVA and Tuckey's multiple comparison test, with differences between groups being considered significant at the $p < 0.05$ level. All statistical tests were conducted using SPSS 25 software.

## Results

### The kid protein showed strong cytotoxicity against sf9 cells

The sf9 insect cell line is commonly used in biological experiments, and typically serves as an insect platform for the production of proteins of interest [20, 21], as well as viruses [22]. Previously, it has been reported that the Kid toxin can induce apoptosis in mammalian cells [14], and therefore we examined whether Kid is also toxic to insect cells. The plasmid that can express Kid and mCherry were constructed. The control plasmid can only express red fluorescent protein. At 48 h post-transfection, expression of red fluorescence in cells transfected with *kid* gene was barely detectable when observed under a fluorescence microscope, with only cell

fragments being mainly observed. However, red fluorescence in cells transfected with control plasmid could be clearly observed after transfection into sf9 cells (Fig 1B and 1C). These observations thereby indicate that the Kid toxin is highly cytotoxic to sf9 cells.

## The presence of tTS did not reduce the background expression level of the Tet-On 3G system in sf9 cells

In the present study, we used the Tet-On 3G system to control expression of the Kid toxin in sf9 cells; however, cell death was still observed even in the absence of doxycycline induction (Fig 2B and 2C, p3), thereby indicating that the background expression level is still too high when this system is used to express highly toxic proteins.

In mammalian cells, the transcription silencing suppressor (tTS) is commonly used to reduce background expression of the Tet-on system, and thus in the present study, we introduced tTS to reduce the leaky expression of Kid driven by the Tet-On system. Nevertheless, following transfection into sf9 cells, we still observed high cell death rates, even in the absence of doxycycline induction (Fig 2B and 2C, p4), and we detected no significant effect of tTS incorporation with respect to controlling the background expression of Kid and levels of cell death (comparing p4 with p3 in Fig 2B). We accordingly deduced that tTS does not achieve the desired effect of controlling background expression in sf9 cells, and thus sought to identify other strategies that could be deployed to reduce the background expression level of the Tet-On 3G system.

## Reducing the background expression level of the Tet-On 3G system using the Csy4 system

Given that, in contrast to mammalian cell, the tTS was ineffective in controlling background expression levels of the Tet-On 3G system in sf9 cells, we needed to develop alternative strategies to reduce background expression. Previous approaches used to regulate gene expression have primarily focused on transcriptional-level regulation. However, we speculated that post-transcriptional regulation, such as mRNA stability management, might be a promising approach to reduce the background expression level of the Tet-On 3G system. Accordingly, in the present study, we investigated the utility of the sequence-specific endoribonuclease Csy4 as a mediator of post-transcriptional regulation. To this end, we inserted the recognition sequence of Csy4 in the 5' UTR region of the target gene, and used an alternative promoter to control the expression of Csy4. On the basis of previous studies, we had established that excessive expression of Csy4 is toxic to sf9 cells, and thus in order to identify suitable promoters for Csy4 expression, we constructed plasmids containing different promoters for controlling *csy4* expression and conducted luciferase activity analyses.

For this purpose, we selected Acie1, WSSVie1 (Δ23) [23], and CMV promoters to control the expression of Csy4 due to their lower activities in insect cells. As shown in Fig 3, the original Tet-On 3G system had a high background expression level, with relative luciferase activities of 100% and 5171 ± 929.8% without and with doxycycline induction, respectively, representing a 51.7-fold induction effect. When tTS was incorporated into the system to control the leaky expression, the relative transcript level reduced to 60.4% and the relative luciferase activity reduced to 56.6% in the absence of doxycycline induction. However, for the expression of highly toxic proteins, this level of background expression is still too high to be acceptable. By replacing the tTS with Csy4, we observed that the background expression level of the target gene was reduced, and in response to an increase in promoter activity in sf9 cells for Csy4 expression, the leaky expression of the target gene was also reduced, for CMV, wssvie1 (Δ23) and acie1 promoters, the relative transcript levels were 21.56 ± 5.65%3.71 ± 0.88%

1.76 ± 0.10%, the relative luciferase activities were 19.32 ± 2.06%, 0.71 ± 0.18%, 0.45 ± 0.20%, respectively. Nevertheless, the maximum expression level after doxycycline induction also decreased correspondingly. Accordingly, we found that all three of the promoters we assessed for driving Csy4 expression were either too weak to decrease the background expression of the target gene (CMV), or too strong, such that excess Csy4 expression markedly affected post-induction expression of the target gene [WSSVie1 (Δ23), and Acie1]. Although at this point, we could have continued to screen for suitable promoters for Csy4 expression, we considered that such a strategy would have taken too much time, and thus we adopted an alternative approach to overcome this problem.

The paradox regarding the use of Csy4 is that a certain level of this enzyme is required before dox-induction to reduce leaky expression, whereas the absence of Csy4 is required for maximal expression upon dox-induction. Accordingly, to remove needless Csy4 following the addition of doxycycline, we inserted a TRE3G promoter-driven reverse *csy4* expression cassette into p8 (Fig 3A) to obtain p10. Our rationale in this regard is that high levels of anti-sense *csy4* mRNA produced via the TRE3G-reverse *csy4* cassette after doxycycline addition would form dsRNA with sense *csy4* mRNA produced by the WSSVie1 (Δ23)-Csy4 cassette, and therefore knock down Csy4 expression [24], and the anti-sense csy4 was not toxic to sf9 insect cells (data not shown), such that the induction of target gene expression would not be affected by Csy4-mediated digestion.

We found that although the production of anti-sense *csy4* mRNA resulted in a moderate increase in target gene background expression, as indicated by a change in relative luciferase activity from 0.71 ± 0.18% to 3.80 ± 0.78%, there was a considerably more pronounced increase in the induced expression of the target gene, with the relative luciferase activity changing from 17.80 ± 2.62% to 1541.80 ± 245.79%, and with a corresponding fold induction increase from 25.0 to 406.0 (Fig 3D). Compared with the induction obtained using p5 and p6, the maximum induced expression decreased to 30% and 18%, respectively; however, the fold increase in induction obtained using p10 was significantly higher than that obtained with either p5 or p6. The 406.0-fold increase in expression thus obtained would satisfy the requirements of most experiments, and the observed reduction in maximum expression could be compensated by transfecting with larger amounts of plasmid.

### The Tet-On 3G-Csy4 system tightly regulated toxin expression

As noted previously, when using either the Tet-On 3G, or Tet-On 3G-tTS system, high levels of leaky Kid expression caused significant cell death (Fig 2B and 2C), indicating an insufficient control of target gene expression in both systems. However, in the system characterized by WSSVie1 (Δ23)-driven *csy4* and TRE3G-driven anti-sense *csy4*, the background expression of *kid* is tightly controlled, without impairing the induction of expression, as little cell death was observed in the absence of doxycycline induction, whereas substantial cell death was detected in the presence of doxycycline (Fig 4B and 4C). These observations thus indicate that our modified Tet-On 3G-Csy4 system is more suitable for driving the inducible expression of toxin genes.

### Discussion

In this study, we found the Kid protein to be highly toxic to insect sf9 cells, and when we attempted to express Kid using the Tet-On 3G system, we observed that leaky expression of the Kid protein was sufficient to kill a proportion of the sf9 cells in the absence of the induction agent doxycycline. The Tet-On 3G system is the third-generation version of the Tet-On system, which is widely used in various fields, including insect expression systems. However, in a

number of circumstances, strictly controlled gene expression is required, and even a small amount of background expression may have a considerable impact on the experimental results. Hence, when using the Tet-On system, lower levels of background expression are required.

In previous studies, the combined use of the Tet-On and tTS systems in HeLa cells has been shown to result in a significant (approximately six-fold) reduction in background expression [12]. However, although the tTS system appears to function efficiently in mammalian cells [11, 25], it seems to operate less well in insects, at least in sf9 cells, for which we found that the combination of tTS and Tet-On 3G did not have the requisite inhibitory effect in terms of reducing leaky target gene expression. The transcriptional inhibition of tTS (TetR-KRAB) in HeLa cells may mediated by binding to the 110 kD SMP-1 (silencing-mediating protein 1) protein [11]. However, whether the lower inhibitory efficiency of tTS in sf9 cells is due to the lack of an insect homolog of SMP1, low levels of SMP1 protein in sf9 cells, or weak interaction between tTS and an insect SMP1, requires further analysis.

Previous studies that have investigated the use of the Tet-On system have tended to focus on the regulation of transcription to reduce the background expression of this system, such as by modifying TRE promoter activity [26], screening rtTA protein mutants to render them more susceptible to doxycycline or using tetracycline transcription-regulated silencers (tTSs) [7, 25]. In contrast to these studies, in the present study, we sought to control leaky expression via post-transcriptional regulation mediated by the sequence-specific Csy4 endonuclease, an enzyme previously used for controlling mRNA stability [19]. When the Csy4 recognition sequence was inserted between the TRE3G promoter and target gene, and the expression of Csy4 was controlled by a series of promoters with different activities, we found that Csy4 could suppress the background expression of mRNA of the Tet-On system to different degrees. We observed that activity of the CMV promoter, which is typically strong in mammalian cells, is very weak in insect cells [27], and thus the background expression of luciferase was still notably high, due to the low levels of Csy4 protein produced when expression is driven by the CMV promoter. Conversely, we found the activities of two alternative promoters, Acie1 and WSSVie1 (Δ23), to be too strong, such that there were substantial decreases in the expression of luciferase, both in the absence and presence of doxycycline induction, and there were significant corresponding decreases in fold induction, an effect we had not anticipated. Accordingly, we assume that a promoter with suitably moderate activity may be required to drive the expression of Csy4, not only for equilibrium of the target gene expression of Tet-On 3G before and after doxycycline induction but also for reducing cytotoxicity caused by excess Csy4. However, we reasoned that identifying such a promoter would probably be too time-consuming, and thus adopted an alternative strategy.

In order to knock down the expression of unwanted Csy4 subsequent to the addition of doxycycline, we employed a TRE3G-controlled anti-sense *csy4*, and accordingly found that compared with that with the expression levels obtained using plasmids p6, p7, and p8, the maximum expression level of luciferase upon induction increased significantly in p10, yielding amounts that were not too less than those obtained using p4 and p5. Although the background expression of luciferase also increased in cells transfected with p10, the fold induction was found to be the higher than that obtained using plasmids p4 to p10, which is the most important index for an inducible expression system, given that the background expression itself can be reduced by simply using lower amounts of plasmid. The Kid expression pattern obtained using p11 also confirmed the low leaky and high inducible expression.

Insect cells are often used for exogenous expression of proteins, including highly toxic proteins, and to achieve maximum expression, the general strategy for expressing these toxic proteins is inducible expression using Tet-On/Off systems or similar inducible systems such as

the rapamycin [28], ecdysteroid [29], and RU486 [30] inducible systems. However, the high-level background expression of target genes obtained when using these systems typically hampers efforts to obtain maximum production of highly toxic proteins, because the leaky expression of toxic proteins is sufficient to kill the insect cells prior to the addition of an inducer. In the present study, we succeeded in reducing the background level of target gene expression to 3.8% of the original level by combining the Tet-On 3G system with the Csy4 system, thereby enhancing its utility for the expression of toxic proteins in sf9 cells. Although we examined the efficiency of the newly developed system only in sf9 cells, this system could potentially be applied to other non-insect systems, including mammals, since the Csy4 can work in a broad range of systems. The character of post-transcriptional regulation using the Csy4 system makes it compatible with almost all existing inducible expression systems, which are generally used to regulate the expression of target genes at the transcriptional level.

## Supporting information

**S1 Table. The primers used in this study.**
(XLSX)

**S1 File. All the sequences of the plasmids used in this study.**
(DOCX)

## Acknowledgments

We would like to thank Editage (www.editage.cn) for English language editing.

## Author Contributions

**Conceptualization:** Chaoliang Lei, Zhihui Zhu.

**Methodology:** Yicheng Zhou.

**Supervision:** Chaoliang Lei, Zhihui Zhu.

**Writing – original draft:** Yicheng Zhou.

**Writing – review & editing:** Yicheng Zhou, Chaoliang Lei, Zhihui Zhu.

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
