## [Decision Letter · Decision Letter 0]

24 Jun 2020

PONE-D-20-14738

A low-background Tet-On system based on post-transcriptional regulation using Csy4

PLOS ONE

Dear Dr. Zhu,

Thank you for submitting your manuscript to PLOS ONE. After careful consideration, we feel that it has merit but does not fully meet PLOS ONE’s publication criteria as it currently stands. Therefore, we invite you to submit a revised version of the manuscript that addresses the points raised during the review process.

We look forward to receiving your revised manuscript.

Kind regards,

Omaththage P. Perera, Ph.D.

Academic Editor

PLOS ONE

Journal Requirements:

Additional Editor Comments (if provided): Dear Author,

We have received comments form two reviewers recommending Major Revisions. Please appropriately address all comments by reviewers. Please reference each reviewer's comments in your response and indicate your revisions in the revised manuscript (Track Changes). Thank you!

**Comments to the Author**

1. Is the manuscript technically sound, and do the data support the conclusions?

Reviewer #1: No

Reviewer #2: Partly

2. Has the statistical analysis been performed appropriately and rigorously? 

Reviewer #1: No

Reviewer #2: No

3. Have the authors made all data underlying the findings in their manuscript fully available?

Reviewer #1: Yes

Reviewer #2: Yes

4. Is the manuscript presented in an intelligible fashion and written in standard English?

Reviewer #1: Yes

Reviewer #2: Yes

5. Review Comments to the Author

Reviewer #1: This is a potentially interesting approach to reduce the basal level of transgene expression, a frequent drawback to the use of the tetracycline-inducible system.

The authors are using a Crispr/Cas9-analog, csy4, associated with a csy4 target sequence placed between the tetracycline-responsive elements (TRE) to reduce transcription. As expected both basal and induced levels were reduced.

In order to differentially affect the induced and uninduced levels, the authors have an antisense csy4 sequence under the control of a secondTRE. Thus, doxycycline treatment presumably resulted in the post-transcriptional blockage of csy4 and increase full expression of the transgene.

However, the study as it is presented falls short in the data justifying the interpretation, lacks important controls and the statistical analysis is not correct.

I) No mRNA analysis is provided to support the proposed mechanism.

II) p10 plasmid: no toxicity analysis has been performed. Therefore the hypothesis that kid 1 is the origin of the toxicity observed with p11 in Fig.4 is not proven. The rccsy4 sequence could be toxic as well.

p11 plasmid: a demonstration that kid 1 is expressed and its level of dox induction is missing. Again, the only evidence that kid 1 is expressed is indirect and in the absence of an appropriate control, the authors cannot exclude that rccsy4 is toxic.

III) The statistical analysis is not adequate:

a. Fig.2 and Fig.4 a student T test can be performed for 2 groups. For more than 2 groups, a one-way ANOVA analysis should be performed. In Fig.4 a two-way ANOVA analysis could also be performed to take into account the Dox treatment.

b. In Fig.3 there is no statistical analysis.

IV) the dox dose is provided only in one case: the luciferase assay. This is a low dose (50ng/ml). Has the same dose been used for all other experiments? A dose range optimization would be interesting.

Reviewer #2: The manuscript presents data on the development of a method to suppress leaky expression by the Tet-On 3G regulated promoter. The employment of the Tet-On 3G promoter has great utility in providing an accurately regulated expression system for many applications. However, the authors clearly demonstrate that the promoter system does have low level leaky expression that can interfere with the desired activity of the system. The coupling of a highly lethal product, Kid protein, with the promoter results in high levels of background cell death as a marker to measure the amount of low-level background transcription from the Tet-On 3G system. To reduce the background further, the authors employ an additional level of negative regulation on the Tet-On 3G system by utilizing a TRE/ WSSVie1 (Δ23) promotor-driven sense/anti-sense csy4 regulator that reduced the background induced cell death and increasing the induction increase by nearly 400-fold.

The authors present their data as luciferase assays and death of fluorescent Sf9 cells. However, the authors do not present any RT-qPCR data on the transcript levels that would support the indirect observations. The RT-qPCR data are critical confirming the transcript reductions reported. The authors also do not report on the collateral effects on the no-transformed cells – or even the rate of transfection. The authors should have included data on the number of non-transformed cells present use a nuclear stain (DAPI) to assess the impact of collateral effects and toxicities from the Kid protein release from the dying cells. The comments that the small fluorescent bodies in figures 2 and 4 represent cell death are not sufficiently clear since there are other objects that are of the same size and fluorescence that appear to be healthy cells in the field. This has to be clarified either by a live stain or another method to validate the observations.

The authors are redundant in the wording of the Results and Discussion. Lines 270 and 348 have “time consuming”, lines 274 and 349 have “unwanted Csy4” showing that the same verbiage has been utilized in both sections. Additionally, “harboring” is used in lines 209, 211 and 354 to indicate the presence of a plasmid. “Harboring” means to shelter or hide something which is not the intent of transfecting these plasmids into the Sf9 cells it is to measure activity.

The manuscript presents a useful new tool to achieve the tightest control of the Tet-On 3G system but the authors should present further evidence that this new system regulation is validated at the level they suggest.

6. PLOS authors have the option to publish the peer review history of their article (what does this mean?). If published, this will include your full peer review and any attached files.

Reviewer #1: Yes: Liliane Tenenbaum

Reviewer #2: Yes: Paul D. Shirk

---

## [Author Response · Author response to Decision Letter 0]

15 Oct 2020

The point-by-point responses to the kind reviwers and the nice editor are included in the attached file entitled "response to the reviewers"

---

## [Decision Letter · Decision Letter 1]

12 Nov 2020

PONE-D-20-14738R1

A low-background Tet-On system based on post-transcriptional regulation using Csy4

PLOS ONE

Dear Dr. Zhu,

Thank you for submitting your manuscript to PLOS ONE. After careful consideration, we feel that it has merit but does not fully meet PLOS ONE’s publication criteria as it currently stands. Therefore, we invite you to submit a revised version of the manuscript that addresses the points raised during the review process.

We look forward to receiving your revised manuscript.

Kind regards,

Omaththage P. Perera, Ph.D.

Academic Editor

PLOS ONE

Additional Editor Comments (if provided):

Both reviewers have expressed concerns about the authors not fully addressing the reviewer's comments. Therefore, I recommend major revision and request authors to address the specific issues raised by the reviewers. Below are the concerns raised by the reviewers during the second round of review:

Reviewer 1:

An important concern (question 3b) has not been addressed.

In Fig.3, neither the luciferase mesaures, nor the mRNA quantification have been submitted to a statistical analysis. Regardeless of the type of measure, a relevant statistical analysis is required to validate the conclusion.

In this Figure, data are presented as means +/- standard deviation. It is only mentionned that “Error bars indicate the standard deviation of three replicates.”. What means “three replicates”? Were 3 independent experiments performed or were the measures of on experiment performed in triplicate?

The authors claim that “There is no literature in the related luciferase experiment for statistical analysis of its results” . This is not correct. Among the citations that they provide, Das, et al. 2016 is a review and Loew, et al. 2010 provide a statistical analysis of luciferase measurements. See also Vanrell et al., 2011 {Vanrell, 2011 #592}, and Chtarto et al., 2016 {Chtarto, 2016 #1056} for example.

Reviewer 2:

The authors have made significant improvement in the communication and clarification of their data. However, the authors still need to address the following issues.

The argument that antisense csy4 is not toxic could have been resolved by simply expressing it in tagged cells, however the indirect evidence is acceptable. A qualifying statement about the lack of antisense csy4 toxicity should be included in the results.

The inclusion of the RT-qPCR data in Figure 3 was useful but the statistical analysis of the data was not included. When analyzing the differences, between samples the authors should have applied ANOVA to show that the differences were significant and then complete the analysis using Tuckey’s multiple comparison test to discriminate between the means of selected treatments. Showing only the fold differences between the treatments does not provide assurance of significance.

Similarly, the Tuckey’s MCT should also be used to complete the analysis of the fluorescent cell counts in Figures 2 and 4. The data is such that a statistical analysis is appropriate.

Please address above points and re-submit the manuscript as soon as possible.

Thank you!

Reviewers' comments:

Reviewer's Responses to Questions

**Comments to the Author**

1. If the authors have adequately addressed your comments raised in a previous round of review and you feel that this manuscript is now acceptable for publication, you may indicate that here to bypass the “Comments to the Author” section, enter your conflict of interest statement in the “Confidential to Editor” section, and submit your "Accept" recommendation.

Reviewer #1: (No Response)

Reviewer #2: (No Response)

2. Is the manuscript technically sound, and do the data support the conclusions?

Reviewer #1: No

Reviewer #2: Partly

3. Has the statistical analysis been performed appropriately and rigorously? 

Reviewer #1: No

Reviewer #2: No

4. Have the authors made all data underlying the findings in their manuscript fully available?

Reviewer #1: No

Reviewer #2: Yes

5. Is the manuscript presented in an intelligible fashion and written in standard English?

Reviewer #1: Yes

Reviewer #2: Yes

6. Review Comments to the Author

Reviewer #1: An important concern (question 3b) has not been addressed.

In Fig.3, neither the luciferase mesaures, nor the mRNA quantification have been submitted to a statistical analysis. Regardeless of the type of measure, a relevant statistical analysis is required to validate the conclusion.

In this Figure, data are presented as means +/- standard deviation. It is only mentionned that “Error bars indicate the standard deviation of three replicates.”. What means “three replicates”? Were 3 independent experiments performed or were the measures of on experiment performed in triplicate?

The authors claim that “There is no literature in the related luciferase experiment for statistical analysis of its results” . This is not correct. Among the citations that they provide, Das, et al. 2016 is a review and Loew, et al. 2010 provide a statistical analysis of luciferase measurements. See also Vanrell et al., 2011 {Vanrell, 2011 #592}, and Chtarto et al., 2016 {Chtarto, 2016 #1056} for example.

Reviewer #2: The authors have made significant improvement in the communication and clarification of their data. However, the authors still need to address the following issues.

The argument that antisense csy4 is not toxic could have been resolved by simply expressing it in tagged cells, however the indirect evidence is acceptable. A qualifying statement about the lack of antisense csy4 toxicity should be included in the results.

The inclusion of the RT-qPCR data in Figure 3 was useful but the statistical analysis of the data was not included. When analyzing the differences, between samples the authors should have applied ANOVA to show that the differences were significant and then complete the analysis using Tuckey’s multiple comparison test to discriminate between the means of selected treatments. Showing only the fold differences between the treatments does not provide assurance of significance.

Similarly, the Tuckey’s MCT should also be used to complete the analysis of the fluorescent cell counts in Figures 2 and 4. The data is such that a statistical analysis is appropriate.

7. PLOS authors have the option to publish the peer review history of their article (what does this mean?). If published, this will include your full peer review and any attached files.

Reviewer #1: **Yes: **Liliane Tenenbaum

Reviewer #2: **Yes: **Paul D. Shirk

---

## [Author Response · Author response to Decision Letter 1]

24 Nov 2020

The point-by-point responses to the kind reviwers and the nice editor are included in the attached file entitled "response to the reviewers"

---

## [Editor Report · Decision Letter 2]

16 Dec 2020

A low-background Tet-On system based on post-transcriptional regulation using Csy4

PONE-D-20-14738R2

Dear Dr. Zhu,

We’re pleased to inform you that your manuscript has been judged scientifically suitable for publication and will be formally accepted for publication once it meets all outstanding technical requirements.

Kind regards,

Omaththage P. Perera, Ph.D.

Academic Editor

PLOS ONE

Additional Editor Comments (optional):

I believe authors have completed all revisions and I recommend accepting the manuscript.
---

## [Editor Report · Acceptance letter]

18 Dec 2020

PONE-D-20-14738R2 

A low-background Tet-On system based on post-transcriptional regulation using Csy4 

Dear Dr. Zhu:

I'm pleased to inform you that your manuscript has been deemed suitable for publication in PLOS ONE. Congratulations! Your manuscript is now with our production department. 

Kind regards, 

on behalf of

Dr. Omaththage P. Perera 

Academic Editor

PLOS ONE